# Semi-Implicit Denoising Diffusion Models (SIDDMs)

**Yanwu Xu**[1,2]*, **Mingming Gong**[3], **Shaoan Xie**[4], **Wei Wei**[1], **Matthias Grundmann**[1],
**Kayhan Batmanghelich**[2]†, **Tingbo Hou**[1]†

[2]Electrical and Computer Engineering, Boston University,
{yanwuxu,kayhan}@bu.edu
[3]School of Mathematics and Statistics, The University of Melbourne
mingming.gong@unimelb.edu.au
[4]Carnegie Mellon University
shaoan@cmu.edu

## Abstract

Despite the proliferation of generative models, achieving fast sampling during inference without compromising sample diversity and quality remains challenging. Existing models such as Denoising Diffusion Probabilistic Models (DDPM) deliver high-quality, diverse samples but are slowed by an inherently high number of iterative steps. The Denoising Diffusion Generative Adversarial Networks (DDGAN) attempted to circumvent this limitation by integrating a GAN model for larger jumps in the diffusion process. However, DDGAN encountered scalability limitations when applied to large datasets. To address these limitations, we introduce a novel approach that tackles the problem by matching implicit and explicit factors. More specifically, our approach involves utilizing an implicit model to match the marginal distributions of noisy data and the explicit conditional distribution of the forward diffusion. This combination allows us to effectively match the joint denoising distributions. Unlike DDPM but similar to DDGAN, we do not enforce a parametric distribution for the reverse step, enabling us to take large steps during inference. Similar to the DDPM but unlike DDGAN, we take advantage of the exact form of the diffusion process. We demonstrate that our proposed method obtains comparable generative performance to diffusion-based models and vastly superior results to models with a small number of sampling steps. The code is available at https://github.com/xuyanwu/SIDDMs.

## 1 Introduction

Generative models have achieved significant success in various domains such as image generation, video synthesis, audio generation, and point cloud generation [1, 2, 3, 4, 5, 6, 7]. Different types of generative models have been developed to tackle specific challenges. Variational autoencoders (VAEs) [8] provide a variational lower bound for training models with explicit objectives. Generative adversarial networks (GANs) [9] introduce a min-max game framework to implicitly model data distribution and enable one-step generation. Denoising diffusion probabilistic models (DDPMs) [10, 1], also known as score-based generative models, recover the original data distribution through iterative denoising from an initial random Gaussian noise vector. However, these models face a common challenge known as the "TRILEMMA" [11], which involves ensuring high-quality sampling, mode coverage, and fast sampling speed simultaneously. Existing approaches, such as GANs, VAEs, and DDPMs struggle to address all three aspects simultaneously. This paper focuses on tackling this TRILEMMA and developing models capable of effectively modelling large-scale data generation.

---

*Work done as a student researcher of Google. † Equal Contribution.

37th Conference on Neural Information Processing Systems (NeurIPS 2023).

While diffusion models excel in generating high-quality samples compared to VAEs and demonstrate better training convergence than GANs, they typically require thousands of iterative steps to obtain the highest-quality results. These long sampling steps are based on the assumption that the reversed diffusion distribution can be approximated by Gaussian distributions when the noise addition in the forward diffusion process is small. However, if the noise addition is significant, the reversed diffusion distribution becomes a non-Gaussian multimodal distribution [11]. Consequently, reducing the number of sampling steps for faster generation would violate this assumption and introduce bias in the generated samples.

To address this issue, DDGANs [11] use a reformulation of forward diffusion sampling and model the undefined denoising distribution using a conditional GAN. This approach enables faster sampling without compromising the quality of the generated samples. Additionally, DDGANs exhibit improved convergence and stability during training compared to pure GANs. However, DDGANs still face limitations in generating diverse large-scale datasets like ImageNet. We propose a hypothesis that the effectiveness of implicit adversarial learning in capturing the joint distribution of variables at adjacent steps is limited. This limitation arises from the fact that the discriminator needs to operate on the high-dimensional concatenation of adjacent variables, which can be challenging.

In order to achieve fast sampling speed and the ability to generate large-scale datasets, we introduce a novel approach called Semi-Implicit Denoising Diffusion Model (SIDDM). Our model reformulates the denoising distribution of diffusion models and incorporate implicit and explicit training objectives. Specifically, we decompose the denoising distribution into two components: a marginal distribution of noisily sampled data and a conditional forward diffusion distribution. Together, these components jointly formulate the denoising distribution at each diffusion step. Our proposed SIDDMs employ an implicit GAN objective and an explicit L2 reconstruction loss as the final training objectives. The implicit GAN objective is applied to the marginal distribution, while the explicit L2 reconstruction loss is adopted for the conditional distribution, where we name the process of matching conditional distributions as auxiliary forward diffusion $AFD$ in our obejctives. This combination ensures superior training convergence without introducing additional computational overhead compared to DDGANs. To further enhance the generative quality of our models, we incorporate an Unet-like structure for the discriminator. Additionally, we introduce a new regularization technique that involves an auxiliary denoising task. This regularization method effectively stabilizes the training of the discriminator without incurring any additional computational burden.

In summary, our method offers several key contributions. Firstly, we introduce a novel formulation of the denoising distribution for diffusion models. This formulation incorporates an implicit and an explicit training objectives, enabling fast sampling while maintaining high-generation quality. Lastly, we propose a new regularization method specifically targeting the discriminator. This regularization technique enhances the overall performance of the model, further improving its generative capabilities. Overall, our approach presents a comprehensive solution that addresses the challenges of fast sampling, high generation quality, scalability to large-scale datasets, and improved model performance through proper regularization.

## 2 Background

Diffusion models contain two processes: a forward diffusion process and the reversion process. The forward diffusion gradually generates corrupted data from $x_0 \sim q(x_0)$ via interpolating between the sampled data and the Gaussian noise as follows:

$$q(x_{1:T}|x_0) := \prod_{t=1}^{T} q(x_t|x_{t-1}), \quad q(x_t|x_{t-1}) := \mathcal{N}(x_t; \sqrt{1-\beta_t}x_{t-1}, \beta_t \mathbf{I})$$

$$q(x_t|x_0) = \mathcal{N}(x_t; \sqrt{\bar{\alpha}_t}x_0, (1-\bar{\alpha}_t)\mathbf{I}), \quad \bar{\alpha}_t := \prod_{s=1}^{t}(1-\beta_s), \tag{1}$$

where $T$ denotes the maximum time steps and $\beta_t \in (0,1]$ is the variance schedule. The parameterized reversed diffusion can be formulated correspondingly:

$$p_\theta(x_{0:T}) := p_\theta(x_T)\prod_{t=1}^{T} p_\theta(x_{t-1}|x_t), \quad p_\theta(x_{t-1}|x_t) := \mathcal{N}(x_{t-1}; \mu_\theta(x_t, t), \sigma_t^2 \mathbf{I}), \tag{2}$$

where we can parameterize $p_\theta(x_{t-1}|x_t)$ as Gaussian distribution when the noise addition between each adjacent step is sufficiently small. The denoised function $\mu_\theta$ produces the mean value of the adjacent predictions, while the determination of the variance $\sigma_t$ relies on $\beta_t$. The optimization objective can be written as follows:

$$\mathcal{L} = -\sum_{t>0} \mathbb{E}_{q(x_0)} D_{\text{KL}}(q(x_{t-1}|x_t, x_0)||p_\theta(x_{t-1}|x_t)), \tag{3}$$

which indirectly maximizes the ELBO of the likelihood $p_\theta(x_0)$. When $x_0$ is given, the posterior $q(x_{t-1}|x_t, x_0)$ is Gaussian. Thus, the above objective becomes $L_2$ distance between the sampled $x_{t-1}$ from the posterior and the predicted denoised data. However, if we want to achieve fast sampling with a small number of steps, The assumption that $p_\theta(x_{t-1}|x_t)$ follows Gaussian does not hold [11], and the $L_2$ reconstruction cannot be applied to model the $KL$ divergence.

To tackle this, DDGANs employ an adversarial learning scheme to match the conditional distribution between $q(x_{t-1}|x_t)$ and $p_\theta(x_{t-1}|x_t)$ via a conditional GAN and enable random large noise addition between adjacent diffusion steps for few-steps denoising. Their formulation can be summarized as follows:

$$\min_\theta \sum_{t>0} D_{adv}(q(x_{t-1}|x_t)||p_\theta(x_{t-1}|x_t)), \tag{4}$$

where $D_{adv}$ tries to distinguish the difference between the predicted and sampled denoising distribution, while the predicted model tries to make them less distinguishable. The objective above can be rewritten as the following expectation:

$$\min_{D_\phi} \max_\theta \sum_{t>0} \mathbb{E}_{q(x_0)q(x_{t-1}|x_0)q(x_t|x_{t-1})} \big[ -\log(D_\phi(x_{t-1}, x_t, t)) \big]$$
$$+ \mathbb{E}_{q(x_t)} \mathbb{E}_{p_\theta(x_{t-1}|x_t)} [-\log(1 - D_\phi(x_{t-1}, x_t, t))]. \tag{5}$$

While this formulation allows more flexible modeling of $p_\theta(x_{t-1}|x_t)$, the pure implicit adversarial learning on the concatenation of $x_{t-1}$ and $x_t$ is statistically inefficient, especially when $x_t$ is high-dimensional. We hypothesize that this is a major reason why DDGANs cannot scale up well on the large-scale dataset with more complex data distributions. In contrast, we will explore the inherent structure in the forward difussion process to develop a more efficient semi-implicit method, which is detailed in the following section.

## 3 Semi-Implicit Denoising Diffusion Models

This section will present our proposed semi-implicit denoising diffusion models (SIDDMs). We first discuss how we reformulate the denoising distribution, which enables fast sampling as DDGANs and high-quality generation as DDPMs. Then, we will introduce how we optimize our model in the training time. At last, we introduce a free discriminator regularizer to boost the model performance further. We show the simplified model structure in Figure 1.

### 3.1 Revisiting Denoising Distribution and the Improved Decomposition

Let us reconsider the training objective presented in Equation 5, which can be reformulated as follows:

$$\min_\theta D_{adv}(q(x_{t-1}, x_t)||p_\theta(x_{t-1}|x_t)q(x_t)), \tag{6}$$

where DDGANs' formulation indirectly matches the conditional distribution of $q(x_{t-1}|x_t)$ and $p_\theta(x_{t-1}|x_t)$ via matching the joint distribution between $q(x_{t-1}, x_t)$ and $p_\theta(x_{t-1}|x_t)q(x_t)$. To keep it concise, we use $p_\theta(x_{t-1}, x_t)$ to represent $p_\theta(x_{t-1}|x_t)q(x_t)$ afterwards.

Starting from this, we can factorize the two joint distributions in the reverse direction and get $q(x_{t-1}, x_t) = q(x_t|x_{t-1})q(x_{t-1})$ and $p_\theta(x_{t-1}, x_t) = p_\theta(x_t|x_{t-1})p_\theta(x_{t-1})$. The conditional distributions are forward diffusion; we name them auxiliary forward diffusion (AFD) in our distribution matching objectives. In this decomposition, we have a pair of marginal distributions of denoised data $q(x_{t-1}), p_\theta(x_{t-1})$ and a pair of conditional distribution $q(x_t|x_{t-1}), p_\theta(x_t|x_{t-1})$. Because the marginal distributions do not have explicit forms, we can match them implicitly by minimizing

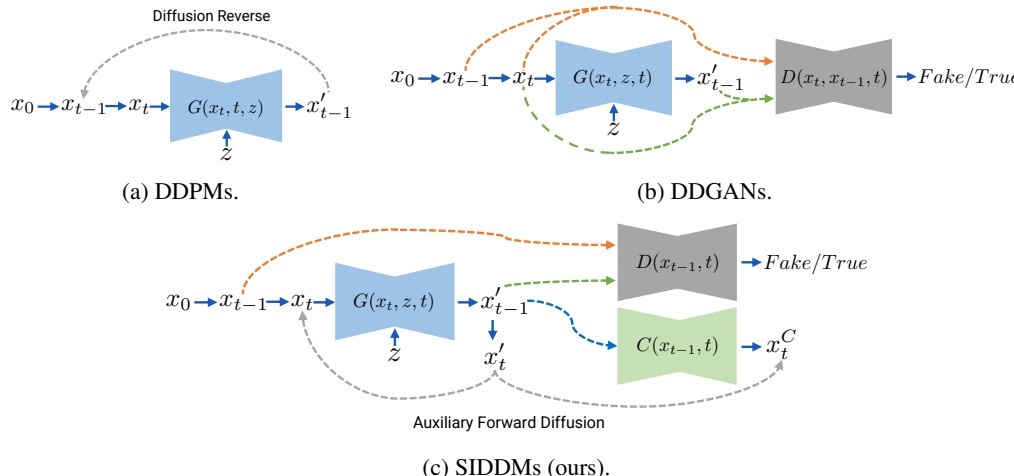

(a) DDPMs.

(b) DDGANs.

(c) SIDDMs (ours).

Figure 1: In this figure, we show three different diffusion models, which are DDPMs, DDGANs and our SIDDMs. These models share some common structures and diffusion processes. Our model shows an improved decomposition of the denoising distribution with the adversarial marginal matching and the auxiliary forward diffusion (AFD) for conditional matching.

the Jensen–Shannon divergence ($JSD$) via adversarial learning. For the conditional distribution of forward diffusion, since $q(x_t|x_{t-1})$ has an explicit form of Gaussian, we can match them via $KL$. The following theorem states that matching these two pairs of distributions separately can approximately match the joint distribution.

**Theorem 1** *Let $q(x_{t-1}, x_t)$ and $p_\theta(x_{t-1}, x_t)$ denote the data distribution from forward diffusion and the denoising distribution specified by the denoiser $G_\theta$, respectively, and we have the following inequality:*

$$JSD(q(x_{t-1}, x_t), p_\theta(x_{t-1}, x_t)) \leq 2c_1 \sqrt{2JSD(q(x_{t-1}), p_\theta(x_{t-1}))}$$
$$+ 2c_2 \sqrt{2D_{KL}(p_\theta(x_t|x_{t-1})||q(x_t|x_{t-1}))},$$

where $c_1$ and $c_2$ are upper bounds of $\frac{1}{2} \int |q(x_t|x_{t-1})|\mu(x_{t-1}, x_t)$ and $\frac{1}{2} \int |p(x_{t-1})|\mu(x_{t-1})$ ($\mu$ is a $\sigma$-finite measure), respectively. A proof of Theorem 1 is provided in Section II of the Supplementary.

### 3.2 Semi-Implicit Objective

Based on the above analysis, we formulate our SIDDMs distribution matching objectives as follows:

$$\min_\theta D_{adv}(q(x_{t-1})||p_\theta(x_{t-1})) + \lambda_{AFD}D_{KL}(p_\theta(x_t|x_{t-1})||q(x_t|x_{t-1})), \quad (7)$$

where the $\lambda_{AFD}$ is the weight for the matching of AFD. In Equation 7, the adversarial part $D_{adv}$ is the standard GAN objective. To match the distributions of AFD via KL, we can expand it as:

$$D_{KL}(p_\theta(x_t|x_{t-1})||q(x_t|x_{t-1})) = \int p_\theta(x_t, x_{t-1}) \log p_\theta(x_t|x_{t-1}) - \int p_\theta(x_t, x_{t-1}) \log q(x_t|x_{t-1})$$
$$= -H(p_\theta(x_t|x_{t-1})) + H(p_\theta(x_t|x_{t-1}), q(x_t|x_{t-1})), \quad (8)$$

which is the combination of the negative entropy of $p_\theta(x_t|x_{t-1})$ and the cross entropy between $p_\theta(x_t|x_{t-1})$ and $q(x_t|x_{t-1})$. In our scenario, optimizing the cross-entropy term is straightforward because we can easily represent the cross-entropy between the empirical and Gaussian distributions using mean square error. This is possible because the forward diffusion, denoted as $q(x_t|x_{t-1})$, follows a Gaussian distribution. However, the negative entropy term $-H(p_\theta(x_t|x_{t-1}))$ is intractable. However, $p_\theta(x_t|x_{t-1})$ can be estimated on samples from the denoiser $G$ that models $p_\theta(x_{t-1}|x_t)$. Thus we need another parametric distribution $p_\psi(x_t|x_{t-1})$ to approximately compute $-H(p_\theta(x_t|x_{t-1}))$.

In the following formulation, we show the maximizing of the conditional entropy can be approximated by the following adversarial training objective:

$$\min_\theta \max_\psi \mathbb{E}_{p_\theta(x_t, x_{t-1})} \log p_\psi(x_t | x_{t-1}). \tag{9}$$

Given $\theta^i$ at the $i$-th iteration, the max step estimates the conditional distribution $p_{\theta^i}(x_t | x_{t-1})$ and we have $p_{\psi^i}(x_t | x_{t-1}) \approx p_{\theta^i}(x_t | x_{t-1})$. Then, in the next iteration, the min step updates the parameter $\theta$ in the generator given $\psi^i$ and obtain the updated $\theta^{i+1}$. Thus, this iterative min-max game between the generator $p_\theta$ and the conditional estimator $p_\psi$ can minimize this negative conditional entropy $-H(p_\theta(x_t | x_{t-1}))$ that we mentioned in the $D_{KL}$ decomposition of Equation 8. This adversarial process can perform as long as we can access the likelihood to $p_\psi(x_t | x_{t-1})$. In our case, it is forward diffusion and follows the Gaussian distribution.

Similar to DDGANs, we also define $p_\theta(x_{t-1} | x_t) := q(x_{t-1} | x_t, x_0 = G_\theta(x_t, t))$ via the posterior distribution. In the distribution matching objective above, we apply the GANs to minimize the $JSD$ of marginal distributions and the $L_2$ reconstruction to optimize the cross entropy. We also define $x'_{t-1}$ as the data sampled from the newly defined distribution, and $x'_t$ are sampled from $x'_{t-1}$ via forward diffusion. Our final training objective can be formulated as follows:

$$\min_\theta \max_{D_\phi, C_\psi} \sum_{t>0} \mathbb{E}_{q(x_0)q(x_{t-1}|x_0)q(x_t|x_{t-1})} \Big[ [\log(D_\phi(x_{t-1}, t))] + [\log(1 - D_\phi(x'_{t-1}, t))]$$

$$+ \lambda_{AFD} \frac{(1 - \beta_t) \left\| x'_{t-1} - x_{t-1} \right\|^2 - \left\| C_\psi(x'_{t-1}) - x'_t \right\|^2}{\beta_t} \Big], \tag{10}$$

where $C_\psi$ denotes the regression model that learns to minimize this negative conditional entropy. In the implementation, we share the most layers between the discriminator and the regression model. We put the detailed derivation of the following training objective in Section I of the supplementary.

Compared with DDGANs, our model abandons the purely adversarial training objective and decomposes it into a marginal and one conditional distribution, where the conditional distribution can be optimised with a less complex training objective and leads to stable training for updating the denoising model. Secondly, our model can share the same model structure as DDPMs and be improved based on the advanced DDPMs network structure. Thirdly, our decomposition does not bring any overhead compared to DDGANs and can achieve steady performance improvement.

### 3.3 Regularizer of Discriminator

The UnetGANs [12] proposes adopting an Unet structure for the discriminator, demonstrating more details in the generated samples. Unlike the common design of discriminators, which only output a global binary logit of "True/Fake", an Unet-like discriminator can distinguish details from different levels. The denoising process in the diffusion models can also benefit from pixel-level distribution matching. Thus, our paper shares the same network structure between the denoiser $G_\theta$ and the discriminator $D_\phi$. Inspired by our decomposition formulation, the reconstruction term provides better gradient estimation and boosts the model performance. We also apply the same strategy to the discriminator as a stand-alone contribution. That is, getting the denoising output from the discriminator and reconstructing it with the ground truth $x_0$. We formulate the regularizer as follows:

$$\min_{D_\phi} \mathbb{E}_{q(x_0)q(x_{t-1}|x_0)} L_2(D_\phi(x_{t-1}, t), x_0), \tag{11}$$

where this regularization only applies to the sampled data $q(x_t)$. Different from the commonly used spectral norm [13], Wasserstein GANs [14] and R1 regularization [15], our regularization does not bring any side effect, such as restriction to model capacity, requiring additional overhead or grid search of the hyper-parameters on each dataset. Our regularizer can be easily plugged into our model and DDGANs and does not require extra network design, which is specifically designed for boosting diffusion models with GANs.

## 4 Related Works

The pioneer works [10, 1] of the diffusion models introduce the discrete time-step diffusion models and the parameterized reversal process for generating novel data. Later, the score-based perspective [16] proposes the continuous-time diffusion models and unifies the denoising diffusion model

and the denoising score matching models [17, 18]. The adversarial score matching [19] utilizes the extra adversarial loss to improve the unimodal Gaussian distribution matching, which differs from the goal of DDGANs and ours. Another branch of works introduces some inductive biases for improving the diffusion models further, the ADM, UDM and EDM [2, 20, 21] propose the classifier guidance, unbounded score matching and better sampling strategy for diffusion-based models respectively.

Although diffusion-based models achieve better quality and diversity, it still suffers from the sampling speed, which usually takes thousands of steps to achieve the best quality. Several methods also boost the sampling speed via knowledge distillation [22, 23], learning the nosing schedule of forward diffusion [24]. The DDIM [25] and FastDPM [26] propose non-Markovian diffusion processes for the sampling boosting. For the score-based continuous-time diffusion models, [27] provides faster SDE solvers. The DDGANs [11] is most related to our proposed methods, which proposed an adversarial formulation for the denoising diffusion distribution and enable a few sampling steps without compromising the generated quality too much.

Unlike DDGANs, we propose a new decomposition for the denoising distribution and achieve better results. From the perspective of better distribution decomposition, we share a similar insight with the related works of conditional GANs (cGANs) [28, 29]. AC-GAN [28] proposes to model the conditional distribution of conditional generative models via decomposing the joint between the label and the data to a marginal and the conditional distribution via an auxiliary classifier. TAC-GAN lately fix the missing term of the AC-GAN decomposition. These two works can only be applied to the conditional generation while our proposed decomposition can be applied to both unconditional and conditional generation. Thus, their works are fundamentally different from ours. The other related GANs propose to enhance GANs with data argumentation [30, 31] or diffusion noise [32], which would also explain why GAN-ehanced denoising diffusion process can function well in practical.

## 5 Experiments

In our experiments, we evaluate our proposed method in the simulated Mixture of Gaussians and several popular public datasets, Cifar10-32 [33], CelebA-HQ-256 [34] and the ImageNet1000-64[35]. To study the effects of our model components, we also conduct the ablations to identify their sensitivity and effectiveness.

For the model architectures, we apply the Unet [36] as the ADM [2], and we follow the same efficient strategy as Imagen [4] to change the order of downsampling and upsampling layer in the Unet blocks. In our method, we also design our discriminator as an Unet. Thus, we apply the identical structure for the generator $G$ and the discriminator $D$. The only difference between them is the input and output channels for fitting in different inputs and outputs in our formulations.

### 5.1 MOG Synthetic Data

Identifying the effectiveness of generative models in high-dimensional data is tricky, and we cannot directly visualize the mode coverage of the generated data. Also, popular metrics like FID and Inception Score can only be referred as quality evaluation. Thus, we test our models and the baseline DDGANs on generating a Mixture of Gaussians. To generate the data, we sample 25 groups of data from the Gaus-

| Model / steps | 1 | 2 | 4 | 8 | 16 |
|---|---|---|---|---|---|
| vanilla GANs | 12.04 | - | - | - | - |
| DDGAN [11] | - | 7.27 | 0.99 | 0.49 | 0.53 |
| SIDDMs(ours) | - | 0.14 | 1.21 | 0.30 | 0.23 |
| SIDDMs w/o AFD (ours) | - | 21.19 | 53.22 | 7.04 | 14.37 |

Table 1: MOG 5x5 results, FID↓

sians independently and each distribution has a different mean but with the same variance. In Figure 3, we show the synthesized results of different models and the denoising steps and the quantitative results are shown in Table 1. We can observe that for the extremely few denoising steps of 2 and many as 16 steps, DDGANs fail to converge to real data distribution, and our proposed method can recover well on the different steps. Also, to identify if the adversarial term takes the main job in the generation, we remove the auxiliary forward term and find that our model cannot recover the original distribution, proving the proposed decomposition's effectiveness.

To evaluate our model's effectiveness on real-world generation tasks, we test our models on the tiny image dataset CIFAR10, the fine-grained generation task on the Celeb-HQ, and the large-scale challenging dataset Imagenet. We pick a small amount of generated images from our model and

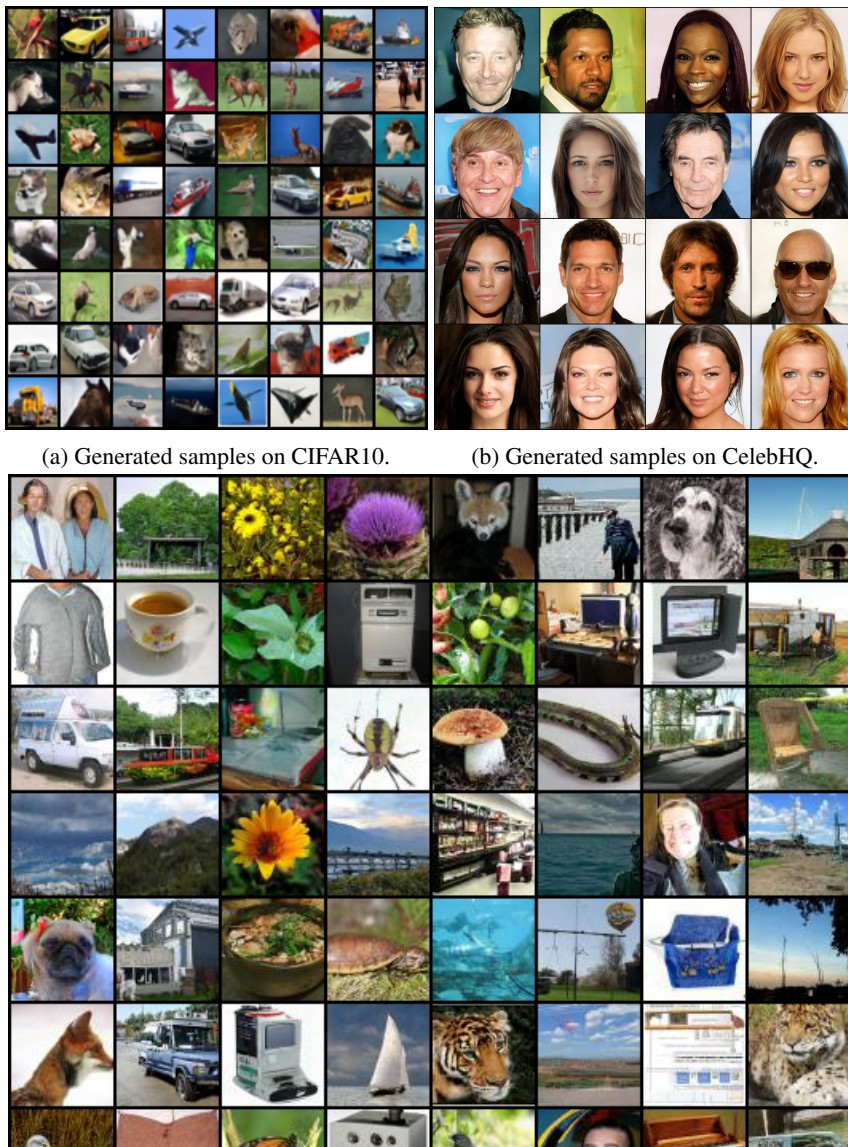

(a) Generated samples on CIFAR10.    (b) Generated samples on CelebHQ.

(c) Generated samples on ImageNet.

Figure 2: We pick a subset of images generated from our model, which produces the paper results. From the top left to the bottom, they are the generated results on CIFAR10, Celeb-HQ and ImageNet.

show them in the Figure 2. We also quantitatively evaluate the quality of the generated samples and collect all of the results in Table 2,3 and 4. Visually, our model generates samples with high fidelity and rich diversity. For the evaluation metric, we choose the Fréchet inception distance (FID) [47] and Inception Score [48] for sample fidelity and the improved recall score [49] to measure the sample diversity. Compared with the GANs-based method, our method achieves better sample quality and diversity, identifying the benefit of enhancing GANs with the diffusion formulation. Compared with the baseline DDGANs, our model shows superior quantitative results than the baseline DDGANs. Although our method does not reach the same quality as the diffusion-based (or score-based) models

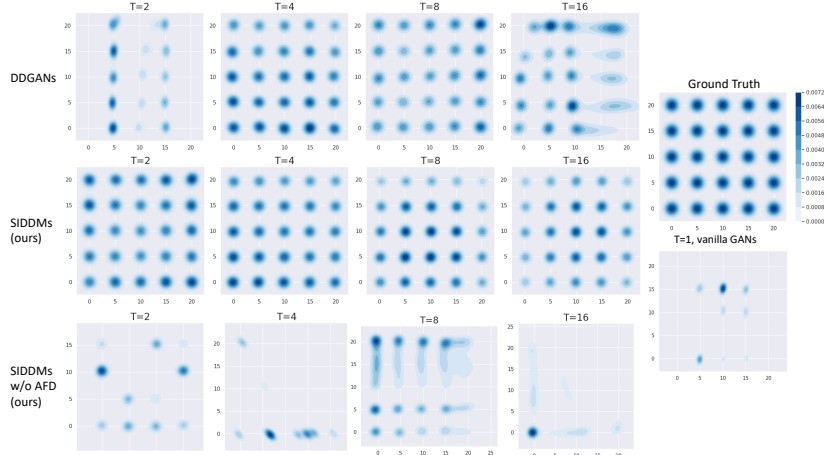

Figure 3: We show the generative results for the $5 \times 5$ Mixture of Gaussians, which can straight-forwardly show the effectiveness of our proposed method. We include our full model, our model without the auxiliary forward diffusion term and the baseline DDGANs. Our model can recover the original distribution even in the small diffusion steps of 2, but the DDGANs fail. Also, without the completed formulation, our model creates biased results.

| Model | IS↑ | FID↓ | Recall↑ | NFE↓ | Time (s)↓ |
|---|---|---|---|---|---|
| SIDDMs (ours), T=4 | 9.85 | 2.24 | 0.61 | 4 | 0.20 |
| DDGANs ([11]), T=4 | 9.63 | 3.75 | 0.57 | 4 | 0.20 |
| Diffusion models | | | | | |
| DDPM [1] | 9.46 | 3.21 | 0.57 | 1000 | 80.5 |
| EDM [21] | 9.84 | 2.04 | - | 36 | |
| Score SDE (VE) [16] | 9.89 | 2.20 | 0.59 | 2000 | 423.2 |
| Score SDE (VP) [16] | 9.68 | 2.41 | 0.59 | 2000 | 421.5 |
| Improved DDPM [37] | - | 2.90 | - | 4000 | - |
| UDM [20] | 10.1 | 2.33 | - | 2000 | - |
| Distillation of Diffusion Models | | | | | |
| DDPM Distillation [38] | 3.00 | - | - | 4 | - |
| CD [22] | 2.93 | 9.75 | - | 2 | - |
| GANs-based models | | | | | |
| StyleGAN2 w/o ADA [31] | 9.18 | 8.32 | 0.41 | 1 | 0.04 |
| StyleGAN2 w/ ADA [31] | 9.83 | 2.92 | 0.49 | 1 | 0.04 |
| StyleGAN2 w/ Diffaug [30] | 9.40 | 5.79 | 0.42 | 1 | 0.04 |

Table 2: Generative results on CIFAR10

| Model | FID↓ |
|---|---|
| SIDDMs (ours) | 7.37 |
| DDGANs ([11]) | 7.64 |
| Score SDE [16] | 7.23 |
| LSGM [39] | 7.22 |
| UDM [20] | 7.16 |
| NVAE [40] | 29.7 |
| VAEBM [41] | 20.4 |
| NCP-VAE [42] | 24.8 |
| PGGAN [43] | 8.03 |
| Adv. LAE [44] | 19.2 |
| VQ-GAN [45] | 10.2 |
| DC-AE [46] | 15.8 |

Table 3: Generative results on CelebA-HQ-256

with a small gap, our model still has the large advantage of fast sampling. To be notified, we apply four denoising steps for the CIFAR10 and two steps for the CelebA-HQ generation, identical to the DDGANs' main results.

## 5.2 Generation on Real Data

For the ImageNet, we choose four denoising steps to get the best sample quality on the ImageNet. ImageNet has 1000 categories, each containing thousands of images and distinguishing itself as a large-scale dataset with diversity. To train a generative model with high fidelity on the dataset, we usually require other inductive biases such as regularizers or model scaling-up methods [13, 50]. The DDGANs fail to retain the high-fidelity samples on this dataset without additional inductive bias. However, with the same model capacity, our proposed method can achieve comparable results w.r.t. the diffusion-based models. This result shows that our model can handle large-scale generation and potentially be applied to boost the sampling speed of large-scale text2image models.

| Model | FID↓ |
|---|---|
| SIDDMs (ours) | 3.13 |
| DDGANs ([11]) | 20.63 |
| CT [22] | 11.10 |
| ADM [2] | 2.07 |
| EDM [21] | 2.44 |
| Improved DDPM [37] | 2.90 |
| CD ([22]) | 4.07 |

Table 4: Generative results on ImageNet1000

# 6 Ablations

Table 5: Study the effect of Auxiliary Forward Diffusion (AFD), FID↓. The weight of 0.0 means that we remove the AFD term and only train the model with adversarial loss. The weight of ∞ means we only keep the AFD term without adversarial loss. This table also evaluates our model performance with different denoising steps and without our regularizer for the discriminator. All of the scores are evaluated on the CIFAR10.

| Weight of AFD | 0.0 | 0.1 | 0.5 | 1.0 | 5.0 | ∞ |
|---|---|---|---|---|---|---|
| SIDDMs(ours) | 77.15 | 3.32 | 2.63 | 2.24 | 2.55 | 41.27 |
| SIDDMs(ours) w/o Regularizer of Discriminator | 51.67 | 4.64 | 4.79 | 3.20 | 2.95 | 51.55 |

| Diffusion Steps | 1 | 2 | 4 | 8 | 16 | 32 |
|---|---|---|---|---|---|---|
| SIDDMs(ours) | 27.39 | 3.17 | 2.24 | 2.31 | 2.15 | 2.46 |

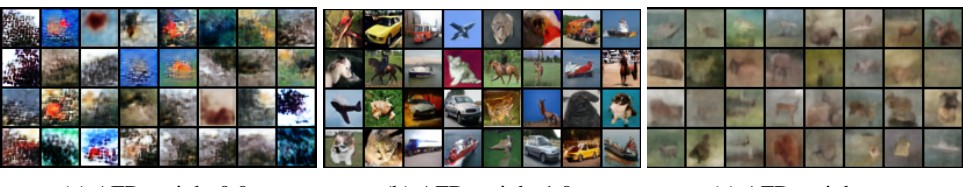

(a) AFD weight 0.0.  (b) AFD weight 1.0.  (c) AFD weight ∞.

Figure 4: Generated samples on CIFAR10, we show two severe cases where only adversarial term (weight 0.0) and only AFD term (weight ∞). We also show our full model results in the middle for comparison.

To identify the function of the components in our formulation, we conduct controlledexperiments on the effects of the adversarial and the AFD term. We set the weights of AFD to be $[0.0, 0.1, 0.5, 1.0, 5.0, \infty]$, where $0.0$ represents only the adversarial term and $\infty$ denotes only the AFD term in our training. We apply the same sampling strategy as our full models and adopt four denoising steps. The FID scores are reported in Table 5, and the generated samples are shown in Figure 4. We can see that if missing any of these two terms in our formulation, we cannot recover the original image distribution. In addition, we found that our model is not sensitive to the weights between these two components w.r.t. the FID scores when both terms participate in the training, which further identifies the effectiveness of our formulation. Also, we train the model without the regularizer for the discriminator, and we also identify that the proposed auxiliary tasks for the discriminator further enhance the model performance for our full formulation.

# 7 Conclusion

In conclusion, the quest for fast sampling with diverse and high-quality samples in generative models continues to pose a significant challenge. Existing models, such as Denoising Diffusion Probabilistic Models (DDPM), encounter limitations due to the inherent slowness associated with their iterative steps. On the other hand, Denoising Diffusion Generative Adversarial Networks (DDGAN) faces scalability issues when dealing with large-scale datasets. To address these challenges, we propose a novel approach that effectively addresses the limitations of previous models by leveraging a combination of implicit and explicit factors. Specifically, we introduce an implicit model that enables us to match the marginal distribution of random variables in the reverse diffusion process. Additionally, we model explicit distributions between pairs of variables in reverse steps, which allows us to effectively utilize the Kullback-Leibler (KL) divergence for the reverse distribution. To estimate the negative entropy component, we incorporate a min-max game into our framework. Moreover, we adopt the L2 reconstruction loss to accurately represent the cross-entropy term in the KL divergence. Unlike DDPM but similar to DDGAN, we do not impose a parametric distribution for the reverse step in our approach. This design choice empowers us to take larger steps during the inference process, contributing to enhanced speed and efficiency. Additionally, similar to DDPM, we effectively leverage the exact form of the diffusion process to further improve our model's performance. Our proposed approach exhibits comparable generative performance to DDPM while surpassing models with fewer sampling steps.

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
