# OpenReview forum: "Semi-Implicit Denoising Diffusion Models (SIDDMs)"
_NeurIPS.cc/2023/Conference — NeurIPS 2023 poster_

### Official Review · Reviewer_6duC · 2023-06-27

**Soundness:** 3 good
**Presentation:** 3 good
**Contribution:** 3 good
**Rating:** 6
**Confidence:** 4

**Summary:**

This paper proposes the Semi-Implicit Denoising Diffusion Model (SIDDM) to enable fast sampling while maintaining high-generation quality. Specifically, SIDDM applies an implicit model to match the marginal distributions of the reverse diffusion process. Meanwhile, SIDDM models explicit conditional distribution of the forward diffusion. Furthermore, a regularization method is proposed to enhance the performance. Experiments show that SIDDM has comparable performance to other diffusion models with fewer sampling steps.

**Strengths:**

1. The authors analyse the reasons for the limitations of DDGAN, and decomposite the denoising distribution to improve the training objective. The idea is reasonable.
2. The experiments in the simulated Mixture of Gaussians and several popular public datasets demonstrate the effectiveness of the proposed method.
3. The authors also provide the code for results reproduction, which shows the solidness of the work.

**Weaknesses:**

1. In Fig. 1, there are dashed lines of different colours in Figure 1. But the authors don't give some corresponding explanation.
2. The implementation details are not clarified, like the structure of the regression model, the discriminator regularizer, and the denoiser. Meanwhile, the training settings (e.g., training iterations) are not provided.

**Questions:**

1. In Fig. 3 and Tab.5, the larger step (e.g., 16) performs worse than smaller step sizes (e.g., 2). This is different from the conventional situation in diffusion models. Please give some analysis.
2. It is recommended to give the training and inference strategy (algorithm) of SIDDM.

**Limitations:**

The authors have discussed the societal impact in the supplementary material. It would be better if the authors discuss some limitations.

---

> ### Author Rebuttal · Authors · 2023-08-08
>
> >  Larger step (e.g., 16) performs worse than smaller step sizes.
>
> In the baseline model DDGANs [11], they also observe that increasing the number of diffusion steps degrades the generative quality. The authors of DDGANs hypothesize that increasing the number of diffusion steps needs more capacity for Discriminator as we need a conditional GAN for each denoising step, where Conditional GANs are difficult to be trained when the number of category labels is large. However, the exact reason still remains unknown. According to our simulation results, it is possible that increasing the number of steps can lead to worse results. Fortunately, we observe that with a small number of diffusion steps our method can already obtain high-quality generations. While we agree that a thorough theoretical understanding of this issue is essential, conducting this analysis is non-trivial and we will definitely consider it in the near future.
>
> >  Training and inference strategy (algorithm) of DDGAN.
>
> For the training settings, we reimplement DDGANs with our proposed GAN training structure and found it would not stable during training without R1 constraint, including the time schedule and the network design. For the inference strategy, we follow the original DDGANs implementation with the posterior sampling which conditioned on the previous $x_t$ and the predicted $x'_0$ from the denoisier. To be more specifically, we write the following algorithm for the DDGAN
>
> Training:
>
> 1. Sampling $x_0\sim q(x_0)), t\sim \text{Uniform}(\{0, . . . , T-1\}), \epsilon_t\sim N(0,1), \epsilon_{t+1}\sim N(0,1)$.
> 2. $x_t=\sqrt{\bar \alpha}x_0 + \sqrt{1 - \bar \alpha}\epsilon_t, x_{t+1}=\sqrt{1-\beta_{t+1}}x_t + \sqrt{\beta_{t+1}}\epsilon_{t+1}$.
> 3. $x_0'=G_\theta(x_{t+1},t+1), x_t' \sim q(x_t|x_0',x_{t+1})$, where $q(x_t|x_0',x_{t+1})$ is the posterior sampling from DDPM.
> 4. D step: $\nabla_\phi (-\log(D_\phi(x_{t-1},x_t,t))-\log(1-D_\phi(x_{t-1}',x_t,t)))$
> 5. G step: $\nabla_\theta (-\log(D_\phi(x_{t-1}',x_t,t)))$
> 6. Repeat 1-5 until model converge.
>
> Inference:
> 1. $x_T\sim N$
> 2. for t = T,...,1 do
>
>    $\epsilon \sim N(0,1)$ if $t>1$ else $\epsilon=0$,
>
>    $x_0'=G_\theta(x_t,t)$,
>
>    $x_{t-1}' \sim q(x_{t-1}|x_0',x_t)$
>
>    end for
>
>   return $x_0$,
>
> where we simply denote the posterior sampling of DDPM as $ x_{t-1}' \sim q(x_{t-1}|x_0',x_t)$ due to the mark down rendering issue.

---

> > ### Comment · Reviewer_6duC · 2023-08-11
> > **After rebuttal**
> >
> > Thanks for the rebuttal.
> >
> > First of all, I need to apologize. Due to my mistake, I mistype the method name in **Summary** and **Question-2**. **I have revised my comment.**
> >
> > For Question-2, I actually hope the authors provide the training and inference strategy of this method (SIDDM).
> >
> > I apologize again for the misunderstanding and inconvenience caused to the authors, area chairs, and other reviewers.
> >
> > Back to the rebuttal, the authors provide an explanation for Question-1. It is better if the authors further provide the method's training and inference strategies (algorithms).

---

> > > ### Author Response · Authors · 2023-08-14
> > > **Training and inference strategy of this method (SIDDM)**
> > >
> > > To reviewer 6duC:
> > >
> > > Hi, No worries, we are glad to provide the training and inference strategy of our proposed method.
> > >
> > > Training:
> > >
> > > 1. Sampling $x_0\sim q(x_0)), t-1\sim \text{Uniform}({0, . . . , T-1}), \epsilon_{t-1}\sim N(0,1), \epsilon_t\sim N(0,1)$.
> > >
> > > 2. $x_{t-1}=\sqrt{\bar \alpha}x_0 + \sqrt{1 - \bar \alpha}\epsilon_{t-1}, x_t=\sqrt{1-\beta_t}x_{t-1} + \sqrt{\beta_t}\epsilon_t$.
> > >
> > > 3. $x_0'=G_\theta(x_t,t), x_{t-1}' \sim q(x_{t-1}|x_0',x_t)$, where $q(x_{t-1}|x_0',x_t)$ is the posterior sampling from DDPM,.$x_t'\sim q(x_t|x_{t-1}')$, where q(x_t|x_{t-1}') is the distribution of forward diffusion.
> > >
> > > 4. D step: $\nabla_{\phi,\psi} (-\log(D_\phi(x_{t-1},t-1))-\log(1-D_\phi(x_{t-1}',t-1))$
> > >
> > >     $+ ||C_\psi(x_{t-1}', t-1))-x_t'||_2)  $
> > >
> > > 5. G step: $\nabla_\theta (-\log(D_\phi(x_{t-1},t-1)) + ||x_t-x_t'||_2$
> > >
> > >     $-||C_\psi(x_{t-1}', t-1))-x_t'||_2)$
> > >
> > > Repeat 1-5 until model converge.
> > >
> > > Our inference follows the same strategy as DDGAN.
> > >
> > > Inference:
> > >
> > > 1. $x_T\sim N(0, 1)$
> > >
> > > 2. for t = T,...,1 do
> > >
> > >     $\epsilon \sim N(0,1)$ if $t>1$ else $\epsilon=0$,
> > >
> > >     $x_0'=G_\theta(x_t,t)$,
> > >
> > >     $x_{t-1}' \sim q(x_{t-1}|x_0',x_t)$
> > >
> > >   end for
> > >
> > > return $x_0$.
> > >
> > > Let us know if you have more concerns on our training or inference strategies, we are more than happy to help you address them.

---

> > > > ### Comment · Reviewer_6duC · 2023-08-20
> > > >
> > > > Thanks for your response. I'm happy to increase my score to 6 (weak accept).

---

> > > > > ### Author Response · Authors · 2023-08-20
> > > > >
> > > > > Thanks again for the reviewing!

---

### Official Review · Reviewer_3HLp · 2023-07-06

**Soundness:** 3 good
**Presentation:** 3 good
**Contribution:** 3 good
**Rating:** 5
**Confidence:** 3

**Summary:**

This method propose a way to achieve fast sampling during inference without compromising sample diversity and quality of diffusion model, and shows their effectiveness on both conditional and unconditional generation, qualitatively and quantitatively. A theoretical framework is also proposed which looks reasonable to me.



**Strengths:**

This method basically proves that a very low FID can be obtained with very few sampling steps, compared to ADM and DDGAN. In addition, it claims that it can do both conditional and unconditional generation. Moreover, the theoretical framework looks solid to me, though I am not from the theory field so I cannot comment more on this.

**Weaknesses:**

However, it seems that there are no experiments on conditional generation (both qualitative and quantitative). Perhaps more relevant experiments are welcome to consolidate this submission and prove the effectiveness of the new sampling strategy.

**Questions:**

The overall story looks good to me. I wonder whether you would like to perfect your experiment with more qualitative/quantitative in the conditional setting.

**Limitations:**

As mentioned above, more analysis on the conditional setting is welcome, e.g., using Stable Diffusion as the base model.

---

> ### Author Rebuttal · Authors · 2023-08-08
>
> > Conditional setting
>
> Thanks for the agreement on our work and the suggestions. It is our mistake that we did not highlight our conditional experiments. In fact, our experiments and the comparisons are conditional generative models on the Imagenet1000  and we have some additional preliminary results on the text2image conditional generation on Laion4B of small UNet setting, shown in Figure 6 in the rebuttal pdf.

---

### Official Review · Reviewer_SF8Z · 2023-07-12

**Soundness:** 2 fair
**Presentation:** 2 fair
**Contribution:** 2 fair
**Rating:** 5
**Confidence:** 4

**Summary:**

In this paper, the authors propose to use an implicit model to match the marginal distributions of noisy data and the explicit conditional distribution of the forward diffusion. Specifically, the adversarial loss is applied to the marginal distributions obtained by forward and backward process, and KL loss is used to regularize the different between the conditional distribution. Experiments show that the method works with small inference steps.

**Strengths:**

* Using the adversarial loss is a good idea to learn the conditional distribution of the backward process with large step size.
* The experimental results with small steps seem to be good, both for toy examples and real datasets.

**Weaknesses:**

* Generally, the paper is hard to follow, the formulas should be presented in a better way:
    * How to obtain equation (6) through (5)?
    * Line 129-137 is really difficult to follow, since $q(x_t|x_{t-1})$ is Gaussian distribution, no matter the step size, why not directly use a Gaussian distribution to parameterize $p_\theta(x_t|x_{t-1})$?
    * It problematic to treat equation (11) and (12) as equivalent, since $\psi$ is fixed in (11).

* In the experiments, with smaller steps, the method works better for both the toy and real datasets. It is not reasonable. Any explanation about this phenomenon?

**Questions:**

Please see weakness part.

**Limitations:**

Please see weakness part.

---

> ### Author Rebuttal · Authors · 2023-08-08
>
> Hi, thanks for the questions, we would love to make the details more readable. Also, due to rendering issue, we place it in the rebuttal pdf.
>
> > Line 129-137 is really difficult to follow
>
> We agree with the reviewer that those line are not written well. Here is our new rewrite and we hope it clarifies that paragraph:
>
> There are two terms in Eq 8. The second term, $H(p(x_t|x_{t-1}),q(x_t|x_{t-1}))$ is a cross entropy which is simply reconstruction loss. This loss can be easily computed because the $q$ generates data, and $p$ does the denoising using $G$. The first term, $H(p (x_t | x_{t-1}))$ is challenging to estimate. The challenges stem from two facts. First, we $p (x_t | x_{t-1})$ is unknown in general, and second, estimating entropy entails computing a high-dimensional integral. However, we observe that  *at the convergence*, $p (x_t | x_{t-1})$ is the same $q (x_t | x_{t-1})$ which is a Gaussian distribution, and we only need to estimate its parameters, $\psi$. Our method can be viewed as a continuous generalization of the method proposed in [29].
>
> > Obtaining equation (6) through (5).
>
> Please refer to the Equation (A) in the rebuttal pdf, where we can rewrite Equation 5 as a typical adversarial training objective with non-saturated loss in the middle expectation, and then, it becomes the joint distribution JSD matching under the sampling strategy proposed by DDGANs. Here we have to admit that  Eq (6) has a typo, where the $E_{q(x_0)q(x_{t-1}|x_0)q(x_t|x_{t-1})}$ in the last joint matching equation should not appear in it. Thanks for your carefully checking.
>
> > It problematic to treat equation (11) and (12) as equivalent
>
> We are sorry for the confusion. We re-wrote that part as the Equation (B) in the rebuttal pdf.  Due to markdown rendering issue, we represent $E$ as the expectation symbol. The $H(X)=E_{q(X)}\log q(X)$ typically represents the entropy for variable $X$. And $H(X,Y) = E_{q(Y)}\log q(X)$  denotes the cross-entropy between two variables $X,Y$. In Equation (10), we have the maximization for the negative cross-entropy and optimize $\psi$, which will result in $p_\psi(x_t|x_{t-1})= p_\theta(x_t|x_{t-1})$ at the optimal. In practice, there might be possible approximation error, i.e., $p_\psi(x_t|x_{t-1})\approx p_\theta(x_t|x_{t-1})$. Thus under this condition, if we try to minimize the cross entropy and optimize $\theta$ while fixing $\psi$, we can rewrite the minimization of the cross entropy as Equation (B).
>
> >. Smaller steps works better.
>
> In the baseline model DDGANs [11], they also observe that increasing the number of diffusion steps degrades the generative quality. The authors of DDGANs hypothesize that increasing the number of diffusion steps needs more capacity for Discriminator as we need a conditional GAN for each denoising step, where Conditional GANs are difficult to be trained when the number of category labels is large. However, the exact reason still remains unknown. According to our simulation results, it is possible that increasing the number of steps can lead to worse results. Fortunately, we observe that with a small number of diffusion steps our method can already obtain high-quality generations. While we agree that a thorough theoretical understanding of this issue is essential, conducting this analysis is non-trivial and we will definitely consider it in the near future.

---

> > ### Comment · Reviewer_SF8Z · 2023-08-21
> >
> > Thanks for the rebuttal, the new version of the paper looks better and clearer and I'll change my rating to 5. However, I am still not convinced about why smaller steps get better results and think the authors can do more work on this part.

---

> > > ### Author Response · Authors · 2023-08-21
> > >
> > > Hi,
> > >
> > > Thanks for the thoughtful suggestions, we will make this explanation better and we are trying to analyze the reason behind the influence of training steps in our method.
> > >
> > > Thanks!

---

> ### Author Response · Authors · 2023-08-20
>
> Hi,
>
> We would greatly appreciate knowing if we have successfully addressed your questions. If you have any additional concerns, please don't hesitate to share them with us.

---

### Official Review · Reviewer_g6aX · 2023-07-27

**Soundness:** 3 good
**Presentation:** 3 good
**Contribution:** 2 fair
**Rating:** 5
**Confidence:** 2

**Summary:**

The paper presents a method called Semi-Implicit Denoising Diffusion Model (SIDDM) aimed to accelerate the sampling process, enhance scalability to large datasets, and improve model performance. It improves upon the DDGAN model by reformulating the denoising distribution of diffusion models with explicit and implicit training objectives, which leverages an implicit GAN objective for the marginal distribution and an L2 reconstruction loss for the conditional distribution. To improve generative quality, the authors adopt a U-net-like structure for the discriminator and a new regularization technique involving an auxiliary denoising task. Experiments are conduct on CIFAR-10, CelebA-HQ-256, and ImageNet to demonstrate it effectiveness.

**Strengths:**

(1) The experiment and comparison were comprehensive, and the results were good. Ablation was comprehensive.
(2) The writing has a clear structure (just not sure if there are any issues with the derivation).
(3) The method performs quite well on large datasets like ImageNet, whereas the previous ones didn't do well.

**Weaknesses:**

(1) There lacks a graph showing the FID-sampling speed tradeoff on ImageNet.
(2) In Table 1, the 'SIDDMs w/o AFD (ours)' entry is quite close to DDGAN in terms of setting, but why is there such a big difference in performance?

**Questions:**

Please see the weakness part.

---

> ### Author Rebuttal · Authors · 2023-08-08
>
> 1. FID-sampling speed tradeoff on ImageNet.
> We have shown the results of FID sample steps in Table 7 of the rebuttal pdf.
>
> 2. 'SIDDMs w/o AFD (ours)' has a large gap on the performance compared with DDGANs.
> This is a good question; In Equation 7, our GAN objective models distribution between marginals and the AFD models the matching between conditionals. They together match the joint distribution between $q(x_{t-1}, x_t)$ and $p_{\theta}(x_{t-1}, x_t)$. Matching the distribution of joints can sufficiently guarantee the matching between the conditionals of $q(x_{t-1}|x_t)$ and $p_{\theta}(x_{t-1}|x_t)$. In the DDGANs, they are also modelling the matching between the joint distribution. However, if we abandon the AFD, it results in only matching the marginals between $q(x_{t-1})$ and $p_{\theta}(x_{t-1})$, which will not guarantee the matching between the conditionals of $q(x_{t-1}|x_t)$ and $p_{\theta}(x_{t-1}|x_t)$. Ultimately, 'SIDDMs w/o AFD (ours)' will output biased distribution.

---

> > ### Comment · Reviewer_g6aX · 2023-08-17
> > **Thanks for the reply**
> >
> > NA

---

### Official Review · Reviewer_azcW · 2023-07-27

**Soundness:** 3 good
**Presentation:** 2 fair
**Contribution:** 3 good
**Rating:** 6
**Confidence:** 4

**Summary:**

The paper addresses the issue of making diffusion models faster (faster sampling) while still yielding high quality, diverse samples, from a large phenomenological space (scaling to large datasets). This is achieved by a novel semi-implicit denoising diffusion model (SIDDM). The idea extends the approach followed in DDGANs by reformulating the objective, and a novel decomposition into two components: (a) a pair of marginal distributions over the denoised data (at each step) with an implicit form to be aligned with the JSD via a learnt critic, and (b) a pair of conditional, forward diffusion components having explicit forms to be aligned using the KL divergence. This mixed objective allows better scaling than DDGANs. A novel regularizer is also introduced for the discriminator which allows a more granular distribution matching. A proof of concept validation is performed using a synthetic, Mixture of Gaussians dataset. Ablations demonstrate the benefit of both the innovations.  SIDDM is benchmarked against the art on the CIFAR10, CelebA-HQ-256, and ImageNet datasets demonstrating performance competitive with DDGANs, better scaling to ImageNet, and much higher sampling speeds than classical DDPMs.



**Strengths:**

**Relevance** The paper addresses an important problem – speeding up diffusion based generative image models while not compromising on the image quality, with good scalability to large, complex datasets. This should be of relevance to the community working in these areas but also of interest to a much wider audience.

**Originality** The proposed decomposition is interesting and novel, balancing both a non-parametric, population-level statistical alignment of distributions and a direct (simple, parametric) objective which decomposes into a sample level objective over Auxiliary Forward Diffusion (AFD). It stands to reason that this can have a significant impact on the training of the diffusion reversal DNN.

**Technical Quality**
- The technical approach mostly appears sound. It creates a new, direct and stronger learning signal which augments the critic based learning objective. This can lead to better learning outcomes over more complex distributions representing larger datasets like ImageNet. In addition, regularizing the discriminator via the UnetGAN formulation seems reasonable too.
- The mathematical formalism used in the paper, including the upper bound in Theorem 1, add to the strength of the contribution, making it principled, and beyond the heuristic choices made. I haven’t verified the derivations in the paper entirely but have followed the structure of the derivations and the proof and find them reasonable.

**Experimental Validation** There are several positives in the choices made for evaluation:
- The choice of using a synthetic dataset based on Mixture of Gaussians (MoGs) allows for validating the core idea.
- While the results on CIFAR10 and CelebA-HQ-256 are comparable with DDGANs, the FID scores on ImageNet demonstrate a performance comparable to SOTA diffusion models while DDGANs performance seems to demonstrate a failure to scale to ImageNet. This validates the core motivation of the paper.
- The ablation study, with and without the AFD term, validates the benefit of the decomposition of the training objective that forms the basis of SIDDMs; as well as the benefit due to the UNetGAN regularizer in (14)

**Significance** The approach has the potential for significant impact over an important area of research once the approach has been reproduced and ‘hardened’.

**Weaknesses:**

**Experimental Validation**

(a) The MoG experiment (Section 5.1) demonstrates that SIDDM achieves good results with very small number of time steps, and has better stability than DDGANs.  However,
- It is not discussed why it is unable to converge to the simple MoG data distribution, when the number of time steps increases (similar to DDGAN).
- Table 1 shows the FID score. These trends don’t tally well with the behavior shown in Figure 3. Perhaps using other metrics (JSD, LPIPS, etc.) may explain the results better?
- The implications of the above on the learning on real datasets, and the scalability thereof on larger,  complex datasets like ImageNet is unclear. Such tasks may require more time steps (it is not clear whether this is the case) in which case, convergence to the data distribution becomes an issue.

(b) The results on CelebA-HQ-256 are typically used to show performance on high-resolution data. The images (depictions) of the generated samples, both in the paper and in the supplementary, are too small. Similarly, the ImageNet depictions are very small.

(c) While NFE (number of function evaluations) and the wall-clock time are shared, no details of the hardware are shared.

**Clarity** (l. 230) It is not quite clear why the material on training with real datasets like CIFAR10, Celeb-HQ etc. is in Section 5.1 and not Section 5.2. Kindly fix.

**Reproducibility** Implementation details are missing. The authors also don’t mention that the code will be released. Given this, I suspect it may be hard to reproduce results.

**Discussion of Limitations** No discussion of limitations in the paper.


**Questions:**

Kindly address the following:
- Explain the behavior on the MoG dataset using a larger suite of metrics (see Borji, CVIU 2019; and its 2021 update for example).
- Explain the behavior (non-convergence to data distribution) when using a larger number of steps and the scalability implications.
- Did you investigate the tradeoffs between a smaller step size with larger number of steps, and smaller number of steps with a larger step size? How does one decide this? Does the theoretical formulation lend itself to such a guidance?
-  Kindly share details of the hardware, implementation, and other details which will aid reproducibility.
- Will the code be shared?
- Discuss limitations.


**Limitations:**

Authors don’t address limitations of the paper.

I don’t think there are any direct negative societal implications. Other limitations and opportunities for improvement are addressed in my responses to previous questions.

---

> ### Author Rebuttal · Authors · 2023-08-08
>
> > Explain the behavior on the MoG dataset using a larger suite of metrics
>
> We pick the Unbiased FID mentioned in Borji, CVIU 2021 and MMD metric for our evaluation of the MOG generative quantitation, and the results are shown in Table 6 MMD is a kernel-based nonparametric metric to measure the difference between two distributions. It has nice theoretical properties and is suitable for the MoG data because here we do not need deep networks to learn representations. For the additional results for Unbaised FID and MMD, we found the scores are pretty close and our model still perform overall better than the DDGAN. There are also other advanced metrics in Borji, CVIU 2021, but some are specifically designed for image datasets or conditional generation cases.
>
> > Explain the behavior (non-convergence to data distribution) when using a larger number of steps and the scalability implications
>
> In the baseline model DDGANs [11], they also observe that increasing the number of diffusion steps degrades the generative quality. The authors of DDGANs hypothesize that increasing the number of diffusion steps needs more capacity for Discriminator as we need a conditional GAN for each denoising step, where Conditional GANs are difficult to be trained when the number of category labels is large. However, the exact reason still remains unknown. According to our simulation results, it is possible that increasing the number of steps can lead to worse results. Fortunately, we observe that with a small number of diffusion steps our method can already obtain high-quality generations. In our paper, we train four-step diffusion on the Imagenet1000, and here we additionally show the preliminary results of our SSIDMs on the text-conditional Laion4B dataset with a small UNet. The preliminary results on a small UNet are shown in rebuttal pdf Figure 6.
>
> >  the tradeoffs between a smaller step size with larger number of steps, and smaller number of steps with a larger step size
>
> In DDGANs and our proposed method, we do not have the Gaussian assumption for $p_{\theta}(x_{t-1}|x_t)$. Thus theoretically, the model trained on any step size with the corresponding number of steps would result in the same distribution convergence at the end of training. However, this claim is based on the assumption that GANs would converge to the optimal, which is often empirically impossible. Our choice of step size with its corresponding number of steps mainly depends on the empirical and DDGANs' observations (we show additional results for the sensitivity of steps in the rebuttal pdf Table 7). While we agree that a thorough theoretical understanding of this issue is essential, conducting this analysis is non-trivial and we will definitely consider it in the near future.
>
> > Details of the hardware, implementation, and other details
>
> Our models are trained on the TPUv4 clusters. A single TPUv4 is around $1.2\times$ faster as A100. And the code is implemented with JAX. Our model structure mostly follows the ADM [2] attention UNet which is also followed by IMAGEN. But instead of predicting the noise from our $G$ model, our model directly reconstructed $x_0'$. We have put our simulation code in the supplementary ".zip" file containing our paper's main formulation. We will release the code for public research once our work gets into the final stage.
>
> > Discuss limitations
>
> Our model incorporated with UNet-like discriminator, and the discriminator needs to see both fake and real data. Thus we have at least two times memory usage than the Diffusion model. Also, in the GANs' training, the model contains two stages of training which results in at most half the training speed for each batch iteration compared with DDPM. These facts would lead to more CO2 emission. However, Distillation on the DDPM model would also take the same amount training time as training DDPM. Thus compared with Distillation+DDPM, our model probably cost the same amount energy and time but with less performance degradation.

---

> > ### Comment · Reviewer_azcW · 2023-08-20
> > **Post rebuttal**
> >
> > Thanks for the detailed response. I have no further questions.
> >
> > I have also gone through the remaining reviews and author responses. I will finalize the rating accordingly.

---

### Author Rebuttal · Authors · 2023-08-09

We want to thank all reviewers evaluating the paper and will fully address all the reviewers' concerns. We put additional Tables and Figures in the rebuttal pdf file, please find the corresponding Tables and Figures for our reponses below. If our answers are satisfactory, we would be thankful if you could update your score. Otherwise, we are happy to answer any more questions.

---

### Author Response · Authors · 2023-08-18

Dear reviewers,

Thanks for your thoughtful comments and suggestions on our model. Please further let us know if we have fully addressed your issues. We are more than pleased to follow up!

---

### Decision · Program_Chairs · 2023-09-21

**Decision:**

Accept (poster)

**Comment:**

There is an agreement between the reviewers to accept the paper. I think that the proposed approach is novel and the experimental validation is sufficient. Therefore, I recommend acceptance.